# Fermented *Aronia melanocarpa* Inhibits Melanogenesis through Dual Mechanisms of the PI3K/AKT/GSK-3β and PKA/CREB Pathways

**DOI:** 10.3390/molecules28072981

**Published:** 2023-03-27

**Authors:** Da Hee Kim, Dong Wook Shin, Beong Ou Lim

**Affiliations:** 1Department of Applied Life Science, Graduate School, BK21 Program, Konkuk University, Chungju 27478, Republic of Korea; 2College of Biomedical and Health Science, Konkuk University, Chungju 27478, Republic of Korea

**Keywords:** *Aronia melanocarpa*, *Monascus purpureus*, fermentation, melanin, anti-melanogenesis, gallic acid

## Abstract

UV light causes excessive oxidative stress and abnormal melanin synthesis, which results in skin hyperpigmentation disorders such as freckles, sunspots, and age spots. Much research has been carried out to discover natural plants for ameliorating these disorders. *Aronia melanocarpa* contains various polyphenolic compounds with antioxidative activities, but its effects on melanogenesis have not been fully elucidated. In this study, we investigated the inhibitory effect of fermented *Aronia melanocarpa* (FA) fermented with *Monascus purpureus* on melanogenesis and its underlying mechanism in the B16F10 melanoma cell line. Our results indicate that FA inhibited tyrosinase activity and melanogenesis in alpha-melanocyte-stimulating hormone (α-MSH)-induced B16F10 cells. FA significantly downregulated the PKA/CREB pathway, resulting in decreased protein levels of tyrosinase, TRP-1, and MITF. FA also inhibited the transcription of MITF by increasing the phosphorylation levels of both GSK3β and AKT. Interestingly, we demonstrated that these results were owing to the significant increase in gallic acid, a phenolic compound of *Aronia melanocarpa* produced after the fermentation of *Monascus purpureus*. Taken together, our research suggests that *Aronia melanocarpa* fermented with *Monascus purpureus* acts as a melanin inhibitor and can be used as a potential cosmetic or therapeutic for improving hyperpigmentation disorders.

## 1. Introduction

Melanin plays a vital role in protecting the skin from ultraviolet light (UV), free radicals, and toxic chemicals in melanocytes. Because repeated ultraviolet exposure increases abnormal reactive oxygen species (ROS) production, our skin protects itself by inducing melanin synthesis [1]. However, the excessive accumulation of melanin causes pigmentation disorders including skin aging, proinflammatory hyperpigmentation, melasma, freckles, age spots, solar lentigo, and even melanoma [2,3,4,5,6]. Melanin is synthesized in the melanosomes, specialized organelles of the melanocytes. This melanogenesis consists of a series of complex processes that are regulated by a variety of factors including enzymes, proteins, and hormones. Tyrosinase is an important enzyme for melanogenesis, and together with tyrosinase-related proteins 1 and 2 (TRP-1 and TRP-2), catalyzes the rate-limiting step of melanin synthesis [7,8]. 

Microphthalmia-associated transcription factor (MITF) is a transcription factor that regulates the expression levels of tyrosinase and is a major regulator of melanogenesis [3]. The alpha-melanocyte-stimulating hormone (α-MSH), which is released from keratinocytes in response to UV irradiation, binds to the melanocortin-1 receptor [9,10]. Subsequently, cyclic adenosine monophosphate (cAMP) activates the protein kinase (PKA) and cAMP response element-binding protein (CREB) [9]. Phosphorylated CREB promotes melanogenesis by directly inducing the transcription of MITF [10]. The activity of MITF is dependent on the extracellular signal-regulated kinase (ERK). The activation of ERK induces MITF phosphorylation at ser73 and, subsequently, MITF ubiquitination and degradation [11,12]. The phosphatidylinositol 3-kinase (PI3K)/AKT signaling pathway is also known to be involved in the regulation of melanin synthesis. PI3K/AKT and glycogen synthase kinase-3 beta (GSK3β) negatively regulate the activity of MITF, thereby suppressing melanogenesis [13]. 

Much research has been actively conducted to find natural products that inhibit melanogenesis [14,15,16]. *Aronia melanocarpa* contains proanthocyanidins, anthocyanins, and phenolic acid contents [17]. It has been shown that the high antioxidant and biological activity of *Aronia melanocarpa* extract results from the presence of active phenolic compounds, which occur indefinitely in higher amounts than in other fruits. Therefore, many studies on this extract are being actively conducted, such as studies on the anti-inflammatory action of *Aronia melanocarpa*, and its ability to lower blood pressure and control cholesterol, but little has been revealed about its effect on melanogenesis [18,19,20,21].

Previous studies have demonstrated that many phenolic acids and flavonoids are effective at inhibiting melanogenesis [22,23,24]. Recent studies have shown that *Monascus purpureus* fermentation can increase the content of polyphenols or flavonoids in the fermentation substrate [25,26,27]. The genus *Monascus* contains three well-known species (*M. purpureus, M. ruber,* and *M. pilosus*), which are often used for rice fermentation to produce red yeast rice [28]. In our study, we fermented *Aronia melanocarpa* with *Monascus purpureus*. We demonstrated a novel approach to elucidate the anti-melanogenic properties of fermented *Aronia melanocarpa* through the regulation of the PKA/CREB and PI3K/AKT/GSK-3β signaling pathways in B16F10 cells.

## 2. Results

### 2.1. Cell Viability of B16F10 Cells with Unfermented Aronia Melanocarpa (UA) and Fermented Aronia melanocarpa (FA)

To examine the cytotoxicity of UA and FA in B16F10 cells, we performed an MTT assay. B16F10 cells were treated with UA and FA at various concentrations for 2 days. In comparison with the control, cell viability was not significantly altered by UA and FA concentrations of up to 800 μg/mL (Figure 1A,B), indicating that UA and FA concentrations below 800 μg/mL were nontoxic to B16F10 cells. Cell viability was above 80% at 50–800 μg/mL. Thus, for subsequent experiments, cells were treated with UA and FA at 500 μg/mL. 

### 2.2. UA and FA Inhibited Melanin Biosynthesis in B16F10 Cells

To investigate the effect of UA and FA on melanin synthesis, the intracellular melanin content was evaluated. B16F10 cells were stimulated with α-MSH (500 nM) [29], and then were treated with UA and FA.

The inhibitory effect of UA and FA on melanin synthesis was compared with that of arbutin (1 mM), a positive control. We found that both UA and FA treatments effectively inhibited α-MSH-induced melanin synthesis (Figure 2). Interestingly, the inhibitory effect of FA on melanin synthesis was significantly stronger than that of UA. 

### 2.3. UA and FA Inhibited Protein Expression Levels of Tyrosinase, TRP-1, TRP-2, and MITF in B16F10 Cells

To examine the effects of UV and FA on melanogenesis-related proteins, a Western blot assay was performed. Protein levels of tyrosinase, TRP-1, TRP-2, and MITF were examined in UA- and FA-treated B16F10 cells with or without α-MSH. As shown in Figure 3A–E, the protein levels of MITF and its downstream tyrosinase families such as tyrosinase, TRP-1, and TRP-2 were significantly decreased upon UA and FA treatments. In addition, FA significantly reduced tyrosinase, TRP-1, TRP-2, and MITF as compared to UA, showing a coincidence with the previous result (Figure 2).

### 2.4. UA and FA Inhibited Melanogenesis by Upregulating p-AKT and p-GSK-3β

In a previous study, the PI3K/AKT/GSK-3β signal pathway was proven to regulate melanogenesis [30]. To understand the underlying mechanism of UA and FA on melanogenesis, the protein expression levels of AKT and GSK-3β were determined. As shown in Figure 4A–C, the α-MSH treatment significantly reduced the expression level of p-GSK-3β in B16F10 cells, and there was no change in the expression of p-AKT. The phosphorylation of AKT and GSK-3β remarkably increased in FA-treated cells as compared to that in the control group and UA group. The FA treatment significantly increased the phosphorylation level of GSK-3β, whereas the UA treatment slightly increased the phosphorylation level of GSK-3β. These results suggest that FA is involved in the PI3K/AKT/GSK-3β signal pathway in the mechanism of melanogenesis inhibition, but UA is not. Therefore, it was found that FA has a stronger melanogenesis inhibitory effect than that UA.

### 2.5. UA and FA Inhibited Melanogenesis by Downregulating p-PKA and p-CREB

In previous research, the signal pathway for MITF-induced melanogenesis was mainly involved with the PKA/CREB axis [10]. To explore the molecular mechanisms underlying the anti-melanogenesis effect of UA and FA, we examined the effects of UA and FA on the expression levels of the melanogenesis-related PKA signaling molecules and CREB proteins by using a Western blot analysis. As shown in Figure 5A–C, the α-MSH treatment significantly increased the phosphorylation level of CREB expression in B16F10 cells but had no effect on the phosphorylation level of PKA. Treatment with 500 μg/mL of FA for 48 h markedly decreased the phosphorylation level of PKA. This condition also significantly reduced the phosphorylation level of CREB. However, in the case of UA, the phosphorylation levels of PKA and CREB were not changed significantly even after treatment under the same condition. These results suggest that FA significantly suppressed melanogenesis through the PKA/CREB pathway. Based on the previous results (Figure 3, Figure 4 and Figure 5), we predicted that a fermentation process could change the content of substances affecting the inhibition of melanogenesis in *Aronia melanocarpa*.

### 2.6. Fermentation of Monascus purpureus Increased Gallic Acid Content in Aronia melanocarpa

Next, we explored which substances in fermented *Aronia* inhibited melanogenesis. A previous study reported that the contents of quercetin and insoluble free phenolic fractions were significantly increased through solid-state fermentation with *Monascus Anka* and *Saccharomyces cerevisiae.* They contained high bioactivity when scavenging for DPPH and ABTS radicals [31]. Additionally, another previous study showed that *Monascus* fermentation could increase the content of polyphenols or flavonoids in the fermentation substrate [25]. The key concept of this study is that *M. purpureus* fermentation can increase the content of polyphenols or flavonoids in the fermentation substrate. Thus, we performed HPLC analysis on both FA and UA. Interestingly, we detected that the content of gallic acid was significantly increased in FA compared to that of UA (Figure 6). Therefore, we expected that the superior efficacy of FA in the inhibition of melanogenesis as compared to UA was due to gallic acid.

### 2.7. Gallic Acid Inhibited Protein Expression Levels of Tyrosinase, TRP-1, TRP-2, and MITF in B16F10 Cells

Previous studies reported that gallic acid inhibited and revealed its underlying molecular mechanism in melanogenesis [32,33]. To investigate whether gallic acid affects the expression levels of tyrosinase, TRP-1, TRP-2, and MITF, a Western blot analysis was used to examine B16F10 cells after treatment with different concentrations of gallic acid (100 μM and 150 μM). As shown in Figure 7A–E, the expression levels of melanogenesis-related proteins were dose-dependently downregulated after B16F10 cells were treated with gallic acid.

The results indicate that the suppressive activity of gallic acid on melanogenesis is linked to the downregulation of MITF and other melanogenesis-related proteins. As a result, it is predicted that the inhibitory activity of tyrosinase, TRP-1, TRP-2, and MITF by FA is superior to that of UA, owing to the increase in gallic acid content through fermentation.

### 2.8. Gallic Acid Inhibited Melanogenesis by Upregulating p-AKT and p-GSK-3β

To understand the underlying mechanism of gallic acid on melanogenesis, the protein expression levels of AKT and GSK-3β were determined. The B16F10 cells were treated with gallic acid (100 μM and 150 μM) and α-MSH for 48 h, and then a Western blot analysis was performed. As shown in Figure 7F–H, treatment with α-MSH significantly decreased the phosphorylated GSK-3β expression in B16F10 cells compared to the control. The phosphorylated AKT expression had no difference compared to the control. When 100-150 μM of gallic acid was treated for 48 h, the phosphorylation levels of AKT and GSK-3β were increased in a concentration-dependent manner. These results suggest that gallic acid inhibited melanogenesis through the PI3K/AKT/GSK3-β pathway, implying that the increase in gallic acid content by *Aronia melanocarpa* fermentation could support the effect of inhibiting melanogenesis.

### 2.9. Gallic Acid Inhibited Melanogenesis by Downregulating p-PKA and p-CREB

To explore the underlying mechanism of gallic acid on melanogenesis, the expression levels of PKA and CREB were determined. The B16F10 cells were treated with gallic acid and α-MSH for 48 h, and a Western blot analysis was performed. As shown in Figure 7I–K, treatment with α-MSH significantly increased the phosphorylation levels of PKA and CREB expression in B16F10 cells compared to the control. Treatment with 100–150 μM of gallic acid for 48 h markedly decreased p-PKA and p-CREB expression levels in a concentration-dependent manner. These results suggest that gallic acid suppresses melanogenesis through the PKA/CREB pathway. Thus, we suggested that FA exhibits an inhibitory effect on melanogenesis owing to the increase in gallic acid content through fermentation.

### 2.10. Inhibition of Melanogenesis by FA Was Associated with PI3K/AKT/GSK-3β Regulation

To determine whether the inhibition of melanogenesis by FA was regulated by the PI3K/AKT/ GSK-3β pathway, B16F10 cells were incubated with a PI3K inhibitor (LY294002) and 500 μg/mL of FA for 48 h. The expression of α-MSH-induced melanogenesis-related proteins was antagonized with an FA treatment.

However, no inhibition of melanogenesis was observed after cotreatment with FA and LY294002 (Figure 8A). Similarly, the expression level of MITF was increased after the α-MSH treatment as compared to the control but was decreased by FA. Additionally, the expression level of MITF was not inhibited when FA and LY294002 were used together as a co-treatment (Figure 8B). Therefore, the cotreatment of FA and LY294002 inhibited the FA-induced upregulation of AKT and GSK-3β phosphorylation, suggesting that PI3K/AKT/GSK-3β is an important signaling pathway involved in mediating the inhibitory effect of FA on melanogenesis (Figure 8).

### 2.11. Inhibition of Melanogenesis by FA Was Associated with PKA/CREB Regulation 

To determine whether the inhibition of melanogenesis by FA is regulated by the PKA/CREB pathway, B16F10 cells were incubated with a PKA inhibitor (H-89) and 500 μg/mL of FA for 48 h. 

Notably, significantly greater inhibition of melanogenesis was observed after co-treatment with FA and H-89 (Figure 9A). Similarly, the expression level of MITF was increased during the α-MSH treatment compared to the control but was decreased by FA. Interestingly, the cotreatment of FA and H-89 exhibited more inhibition compared to the other group (Figure 9B). Therefore, the cotreatment of FA and H-89 significantly suppressed the expression level of MITF compared to FA alone, suggesting that PKA/CREB is an important key signaling pathway involved in mediating the inhibitory effect of FA on melanogenesis (Figure 9).

## 3. Discussion

Melanin, which is the end-product of the complex, multistep transformation of L-tyrosine, is a group of polymorphous and multifunctional biopolymers that are represented by eumelanin, pheomelanin, neuromelanin, and mixed melanin pigment in melanocytes [34,35,36]. Previous studies have suggested that melanin plays an important role in photoprotection and acts as a sunscreen; however, excessive melanin deposition after exposure to UV radiation can lead to a skin hyperpigmentation disorder [37,38,39,40]. Tyrosinase is the main enzyme for melanin synthesis, and its transcription is regulated by MITF. MITF acts as the master regulator of melanocyte development, function, and survival [41]. MITF plays a central role in melanin synthesis, as well as in melanosome biogenesis and transport [42]. The expression level of MITF is regulated by many factors, including CREB, PI3K/AKT, GSK-3β, and MAPKs (ERK, c-Jun N-terminal kinase, and p38) [43,44,45,46]. MITF is thought to mediate significant effects of α-MSH by transcriptionally regulating enzymes that are essential for melanin production, including tyrosinase, TRP-1, and TRP-2 [47]. 

*Aronia melanocarpa*, due to the presence and the high content of bioactive components, exhibits a wide range of positive effects, such as strong antioxidant activity and potential therapeutic benefits [17,18,20,48]. It can also contribute to the prevention of chronic diseases, including metabolic disorders, diabetes, and cardiovascular diseases, because of its supportive impacts on lipid profiles, fasting plasma glucose, and blood pressure levels [19,48,49]. However, little has been reported about the melanogenesis of *Aronia melanocarpa*. Therefore, our work is the first study to report on the inhibitory effect on the melanogenesis of *Aronia melanocarpa*. In this study, *Aronia melanocarpa* was fermented using *M. purpureus.* The genus *Monascus* contains three well-known species (*M. purpureus*, *M. ruber,* and *M. pilosus*) [25]. Recent studies have shown that *Monascus* fermentation can increase the content of polyphenols or flavonoids in the fermentation substrate [25,28]. Based on these studies, we fermented *Aronia* with *M. purpureus*, confirmed the increase in the content of gallic acid, and examined the underlying mechanism of the gallic acid on the inhibition of melanogenesis. Gallic acid is a plant polyphenol antioxidant that has been shown to have various biological properties [32,33]. In the current study, we investigated the molecular mechanism regarding the inhibitory effect of gallic acid on melanogenesis in B16F10 cells. We found that gallic acid significantly suppressed tyrosinase, TRP1, TRP2, and MITF in a dose-dependent manner (Figure 7A). Moreover, the gallic acid treatment upregulated the phosphorylation levels of PI3K/AKT in a dose- and time-dependent manner (Figure 7F), whereas the gallic acid treatment downregulated the phosphorylation level of CREB (Figure 7I). Furthermore, we revealed that FA was dominant regarding the anti-melanogenesis effect over UA in all respects. Therefore, we postulate that the inhibitory effect of FA on melanogenesis was due to gallic acid, which was increased by the fermentation process of *M. purpureus*. 

To understand the mechanism of the anti-melanogenic action of FA, the melanogenesis-related enzymes were evaluated. The expression of tyrosinase, TRP-1, and TRP-2 was decreased by FA (Figure 3A), which was consistent with the results of melanin synthesis. In the study, FA inhibited CREB activation and, consequently, decreased MITF expression in B16F10 cells (Figure 5A). Reduced MITF expression resulted in the downregulation of tyrosinase, TRP-1, and TRP-2 enzymes and the suppression of melanin content. Furthermore, the inhibitory activity of FA against tyrosinase partially contributed to decreasing melanin production (Figure 2). PI3K/AKT/GSK-3β has been recognized as another important signaling cascade that regulates the transcriptional activity of MITF. FA increased the phosphorylation of AKT and GSK-3β, leading to the decreased expression of tyrosinase, TRP-1, and MITF (Figure 4A).

To figure out the molecular mechanism underlying the inhibitory effect of FA on melanogenesis, several PKA- and PI3K-related kinase inhibitors were used. In our study, melanogenesis and the expression level of MITF were inhibited in B16F10 cells after the cells were cotreated with FA and H-89 (Figure 9A). In contrast, melanogenesis and the expression level of MITF were increased in B16F10 cells after the cells were cotreated with FA and LY294002. In our study, FA upregulated the phosphorylation levels of AKT and GSK-3β, decreased MITF transcription, and eventually, suppressed melanogenesis (Figure 8A). The gallic acid content was increased in *Aronia melanocarpa* due to the fermentation of *M. purpureus,* which contributed to inhibiting melanogenesis (Figure 6). In conclusion, FA inhibits melanogenesis by increasing the activity of the AKT/GSK-3β pathway. In addition, FA inhibits the activity of the PKA/CREB pathway, thereby inhibiting melanogenesis. We had a limitation in that our study was performed on murine melanoma and it remains to be clarified whether a similar mechanism may operate in humans. In this study, our results suggest that FA may be used as a healthy functional food or a cosmetic ingredient for improving hyperpigmentation disorders. 

## 4. Materials and Methods

### 4.1. Chemicals and Materials

Dulbecco’s modified Eagle’s medium (DMEM) was purchased from WElGENE Inc. (Republic of Korea). Gallic acid was purchased from Sigma-Aldrich (St. Louis, MO, USA) for use in high-performance liquid chromatography (HPLC) and for investigating the mechanism of gallic acid on melanin synthesis. α-melanocyte-stimulating hormone, arbutin, LY294002, and PD98059 were purchased from Sigma-Aldrich (St. Louis, MO, USA). Antibodies recognizing GAPDH, AKT, phospho-AKT, GSK-3β, phospho-GSK-3β, CREB, phospho-CREB, PKA, and phospho-PKA were obtained from Cell Signaling Technology (Beverly, MA, USA). Other primary antibodies and secondary antibodies were purchased from Santa Cruz Biotechnology, Inc. (Santa Cruz, CA, USA). All other unlabeled chemicals and reagents were analytical grade and purchased from Sigma-Aldrich (St. Louis, MO, USA).

### 4.2. Cell Culture

B16F10, a mouse melanoma cell line, was obtained from the Amorepacific Corporation and maintained in DMEM supplemented with 10% heat-inactivated fetal bovine serum (FBS), 100 U/mL of penicillin, and 100 mg/mL of streptomycin in a humidified atmosphere of 5% CO2 at 37 °C. Cell numbers were assessed using a hemocytometer.

### 4.3. Cytotoxicity Assay

Cell viability was evaluated by the ability of mitochondria to reduce tetrazolium salt (MTT) into MTT formazan crystals. B16F10 cells (2 × 10^4^ cells/well) were seeded in 96-well plates and incubated for 6 h at 37 °C. The cells were treated with different concentrations of the sample for 18 h. After 18 h, the MTT solution (2 mg/mL in PBS) was added to each well and incubated for 4 h at 37 °C. The culture medium was discarded, and DMSO was added to each well to solubilize the formazan. The absorbance of the solubilized formazan solution was then measured at 570 nm.

### 4.4. Preparation of UA and FA

*Aronia melanocarpa* was purchased from Danyang Aronia Farming Association in Chungbuk Province Inc. (Republic of Korea). The Aronia was washed 3 times with distilled water and dried. Dried *Aronia melanocarpa* fruits were ground into a fine powder using a grinder. About 200 g of ground powder was extracted, and this process was repeated 3 times with 2 L of 70% ethanol. Ethanol was evaporated from the extract using a vacuum rotary concentrator. A part of the Aronia extract was used as unfermented Aronia (UA) and the rest was fermented. Then, the Aronia extract was mixed with 5% *Monascus purpureus* in a 1 L flask and fermented at 30 °C for 14 days on a shaker at 150 rpm. The residue after fermentation was purified by filtering at room temperature through Whatman No. 1 filter paper (GE Healthcare UK Ltd., Buckinghamshire, UK). The unfermented *Aronia melanocarpa* (UA) and fermented *Aronia melanocarpa* (FA) were lyophilized at −80 °C, and the samples were stored at −20 °C until used.

### 4.5. High-Performance Liquid Chromatography (HPLC) of Sample 

HPLC analysis of the sample was performed according to the method of [28]. Identification of the polyphenol compounds in the Aronia sample was performed using an HPLC system (Dionex UltiMate 3000 Series, Thermo Fisher Scientific, Waltham, MA, USA) equipped with a Syncronis C18 analytical column (5 μm, 250 mm × 4.6mm) (Thermo Fisher Scientific). The analytical conditions for HPLC included a column temperature of 25 °C, a flow rate of 0.7 mL/min, an injection volume of 10 mL, and a wavelength of 280 nm. Mobile phases were composed of 0.1% trifluoroacetic acid (A) and acetonitrile (B), where the gradients were 15–30% B at 0–20 min, 50% B at 20-75 min, and 30% B at 75–80 min. Peak identification of phenolic acid was confirmed via the retention time and the spectrum of the standards. The phenolic acid standard, including gallic acid, was purchased from Sigma-Aldrich (St. Louis, MO, USA). The data were collected using the Chromeleon Chromatography Management System (version 6.80; Thermo Fisher Scientific).

### 4.6. Measurement of Melanin Content in B16F10 Cells 

The melanin contents of B16F10 cells were measured according to the method of Jeong Y et al. using an enzyme-linked immunosorbent assay (ELISA) reader at 410 nm [29]. Briefly, the B16F10 cells were cultured in 100mm culture plates and incubated. The cells were treated with a medium containing α-MSH and various concentrations of UA, FA, and arbutin for 48 h. PRO-PREP buffer (iNtRON Biotechnology, Seongnam, Korea) was added to each well to lyse the cells, and the cells were then centrifuged. After centrifugation for 3 min at 2000 rpm, the cell pellet was dissolved in 100 μL of a 1 N sodium hydroxide (NaOH) solution containing 10% DMSO for 60 min at 80 °C. The absorbance of the solution was measured at 410 nm using the ELISA microplate reader.

### 4.7. Western Blot Analysis

B16F10 cells were lysed in PRO-PREP buffer (iNtRON Biotechnology, Seongnam, Korea). Total protein concentration was determined using a Bradford assay with bovine serum albumin as the standard. Total protein was separated via sodium dodecyl sulfate-polyacrylamide gel electrophoresis (SDS-PAGE) and was then transferred to polyvinylidene fluoride (PVDF) membranes (Bio-Rad, Hercules, CA, USA). PVDF membranes were blocked overnight with 5% BSA solution and with the following specific antibodies: AKT, phospho-AKT, GSK-3β, phospho-GSK-3β, CREB, phospho-CREB, PKA, phospho-PKA, ERK, phospho-ERK, and GAPDH (Cell Signaling Technology, Beverly, MA, USA); MITF, TRP-1, TRP-2, and tyrosinase (Santa Cruz Biotechnology, Inc., Santa Cruz, CA, USA). The latter group was detected using a chemiluminescence reagent (SurModics IVD, Inc., Eden Prairie, MN, USA), and was then visualized and analyzed using the ChemiDoc XRS+ Imaging System (Bio-Rad, CA, USA) equipped with Image Lab.

### 4.8. Statistical Analysis

Data are presented as the mean ± standard deviation (SD). All the analyses were carried out in triplicate. Statistical analyses were performed by using a one-way analysis of variance (ANOVA). Significant differences between groups were determined at *p* < 0.05. GraphPad Prism 5, Sigma plot 10.0, and Microsoft Excel were used for the statistical and graphical evaluations.

## Figures and Tables

**Figure 1 molecules-28-02981-f001:**
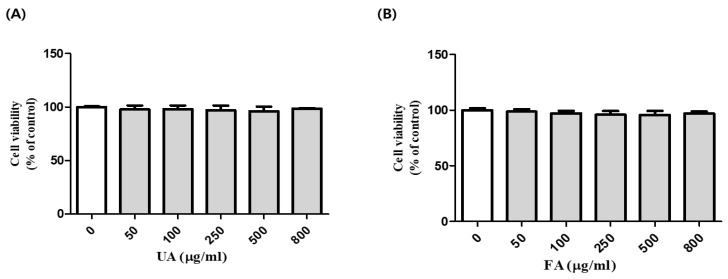
Cell viability (%) of B16F10 cells after 48 h of treatment with UA (**A**) and FA (**B**). Cell viability was not significantly altered by UA and FA concentrations of up to 800 μg/mL. Each value is presented as the mean ± SD.

**Figure 2 molecules-28-02981-f002:**
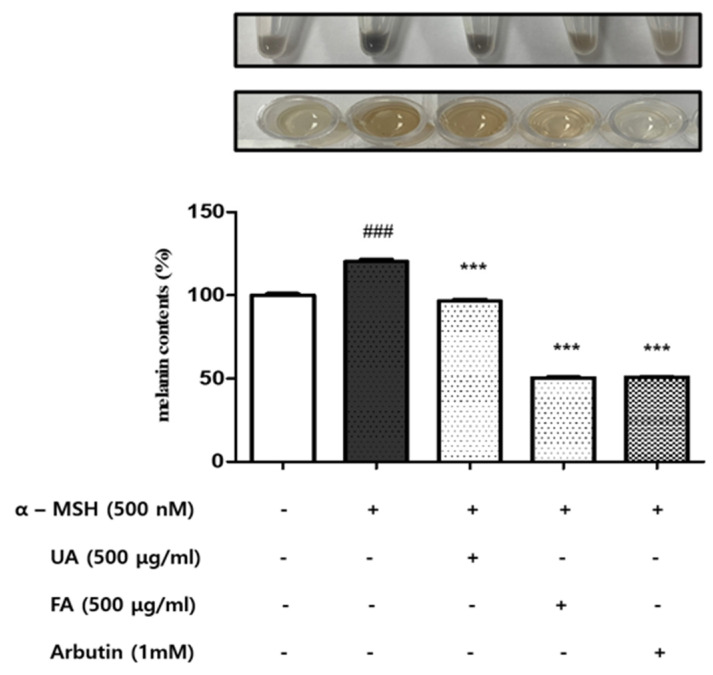
Melanin content (%) of B16F10 cells and cell pellets after 48 h of treatment with UA and FA. FA significantly inhibited melanin synthesis compared to UA. Each value is presented as the mean ± SD. Significant difference versus control: ### *p* < 0.001. Significant difference versus α-MSH-treated group: *** *p* < 0.001. Positive control: 1 mM of arbutin.

**Figure 3 molecules-28-02981-f003:**
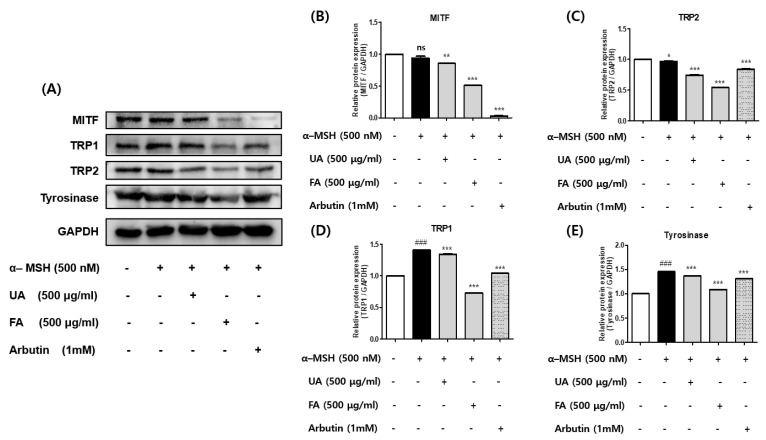
(**A**) Protein levels of tyrosinase, TRP-1, TRP-2, and MITF were examined in UA-and FA-treated B16F10 cells with or without α-MSH. FA significantly inhibited melanin synthesis compared to UA. Each value is presented as the mean ± SD. Significant difference versus control: ### *p* < 0.001. Significant difference versus α-MSH-treated group: * *p* < 0.05, ** *p* < 0.01, *** *p* < 0.001. Positive control: 1 mM of arbutin. (**B**–**E**) Relative expression levels of proteins. Each value was calculated from the ratio of the signal intensity for the indicated protein to that of GAPDH. Data are shown as mean ± standard deviation values from two separate experiments. Each value is presented as the mean ± SD. Significant difference versus control: ### *p* < 0.001. Significant difference versus α-MSH-treated group: * *p* < 0.05, ** *p* < 0.01, *** *p* < 0.001.

**Figure 4 molecules-28-02981-f004:**
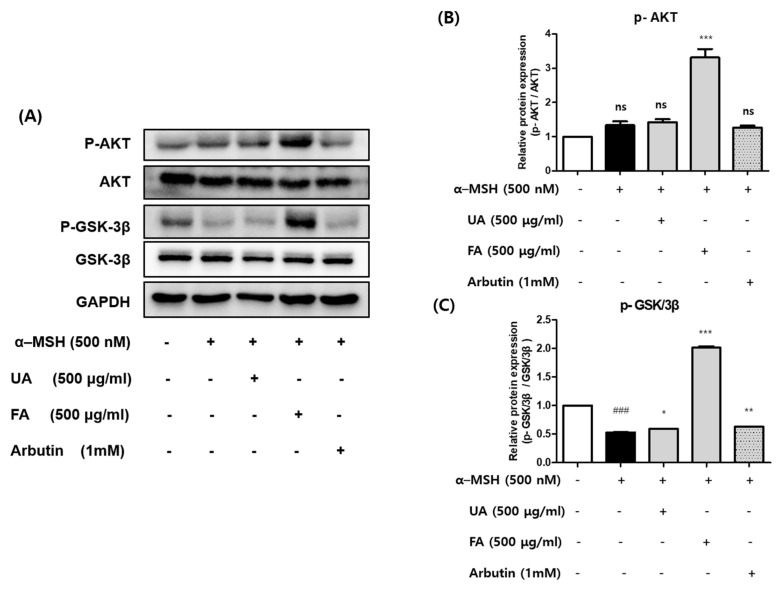
(**A**–**C**) Effect of UA and FA on α-MSH-induced protein expression of p-GSK3β and p-AKT in B16F10 cells. Each value is presented as the mean ± SD: in comparison with the control group, as ### *p* < 0.001; in comparison with the α-MSH-treated group, as **p* < 0.05 ** *p* < 0.01 and *** *p* < 0.001. ns, not significant.

**Figure 5 molecules-28-02981-f005:**
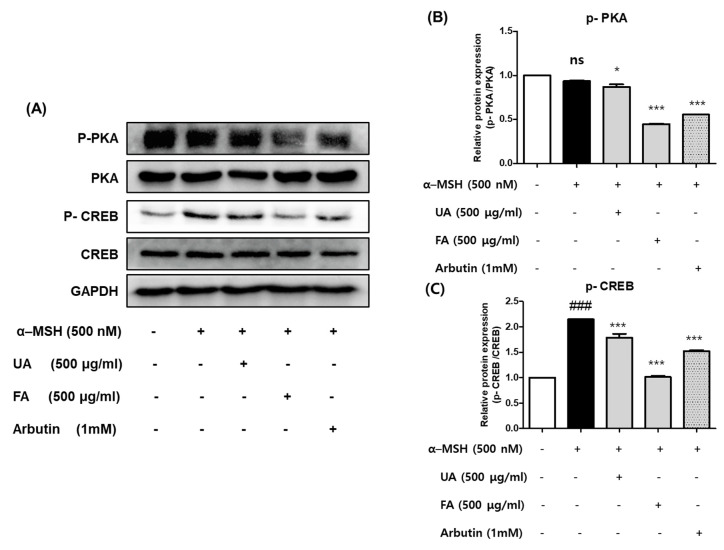
(**A**–**C**) Effect of UA and FA on α-MSH -induced protein expression of p-PKA and p-CREB in B16F10 cells. Each value is presented as the mean ± SD: in comparison with the control group, as ### *p* <0.001; in comparison with the α-MSH-treated group, as * *p* < 0.05 and *** *p* < 0.001. ns, not significant.

**Figure 6 molecules-28-02981-f006:**
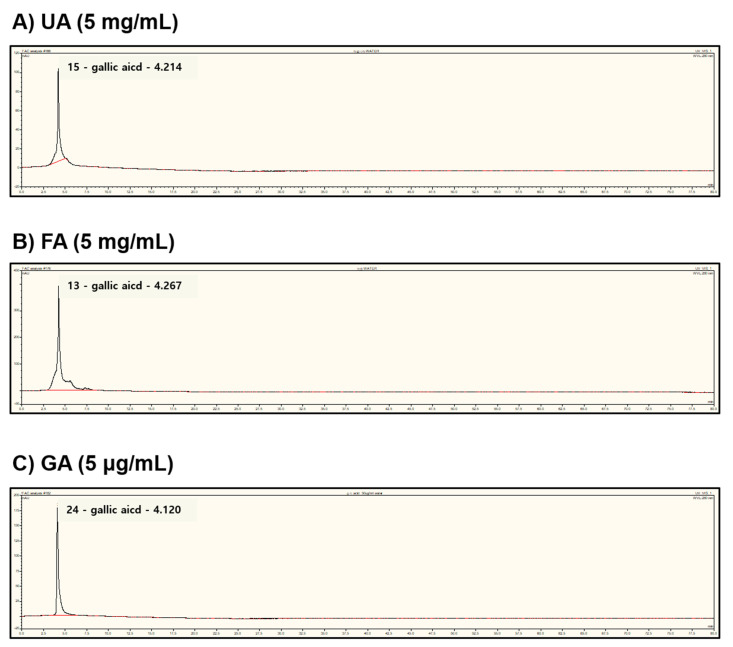
HPLC chromatograms of gallic acid determined with a 10 μL injection of UA and FA. HPLC chromatograms are recorded at 270 nm. The red line was a baseline. (**A**) UA—unfermented Aronia; (**B**) FA—fermented Aronia; (**C**) GA—gallic acid.

**Figure 7 molecules-28-02981-f007:**
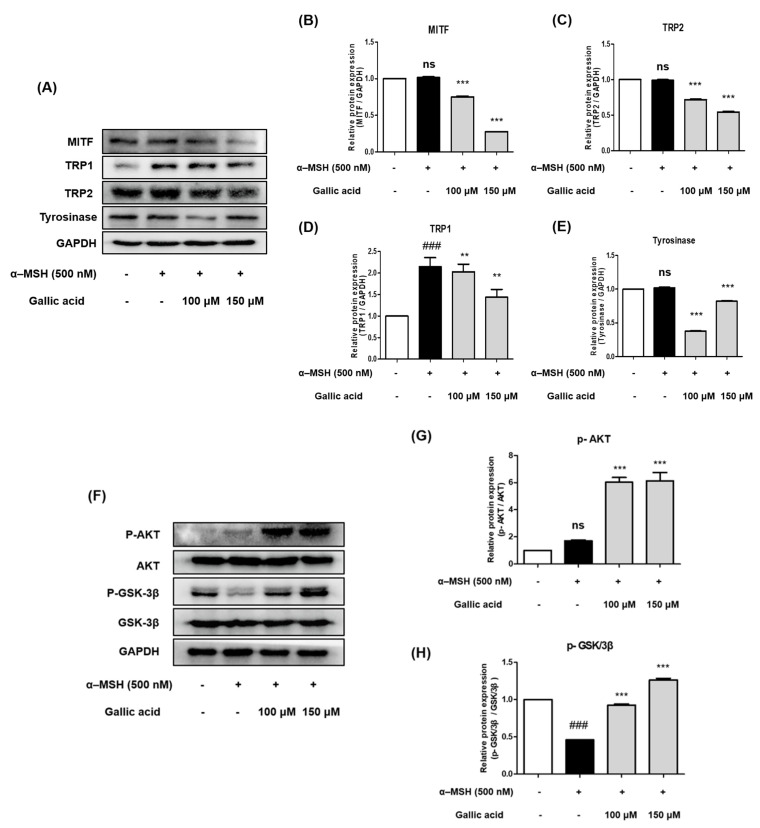
(**A**–**E**) Effect of gallic acid (100uM and 150uM) on α-MSH-induced protein expression of tyrosinase, TRP-1, TRP-2, and MITF in B16F10 cells. Each value is presented as the mean ± SD: in comparison with the control group, as ### *p* <0.001. ns, not significant; in comparison with the α-MSH-treated group, as ** *p* < 0.01 and *** *p* < 0.001. (**F**–**H**) Effect of gallic acid on α-MSH-induced protein expression of p-GSK3β and p-AKT in B16F10 cells. Each value is presented as the mean ± SD: in comparison with the control group, as ### *p* <0.001. ns, not significant; in comparison with the α-MSH-treated group, as ** *p* < 0.01 and *** *p* < 0.001. (**I**–**K**) Effect of gallic acid on α-MSH-induced protein expression of p-PKA and p-CREB in B16F10 cells. Each value is presented as the mean ± SD: in comparison with the control group, as ### *p* <0.001; in comparison with the α-MSH-treated group, as *** *p* < 0.001.

**Figure 8 molecules-28-02981-f008:**
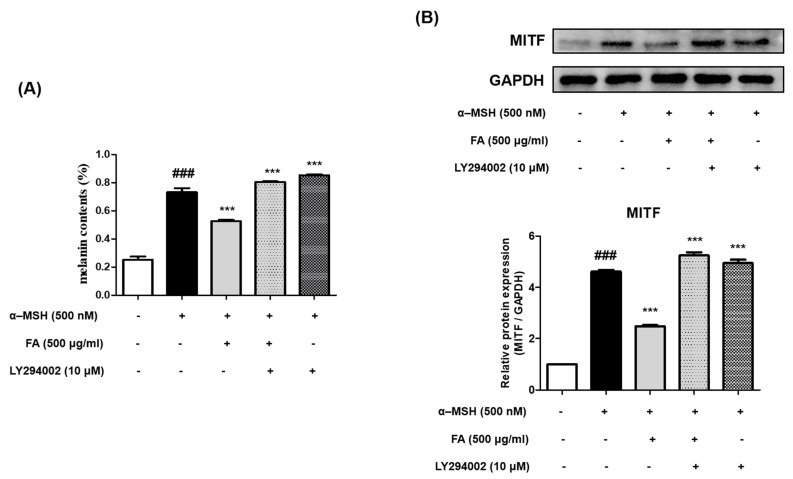
(**A**) Effects of FA and LY 294002 (PI3K inhibitor) on melanin content (%) in α-MSH-treated B16F10 cells after 48 h. Each value is presented as the mean ± SD. ### *p* < 0.001 (significant difference versus control); *** *p* < 0.001 (significant difference versus α-MSH-treated group). (**B**) Effects of FA and LY 294002 (PI3K inhibitor) on protein expression of MITF in α-MSH-treated B16F10 cells after 48 h. Each value is presented as the mean ± SD. *** *p* < 0.001 (significant difference versus α-MSH-treated group).

**Figure 9 molecules-28-02981-f009:**
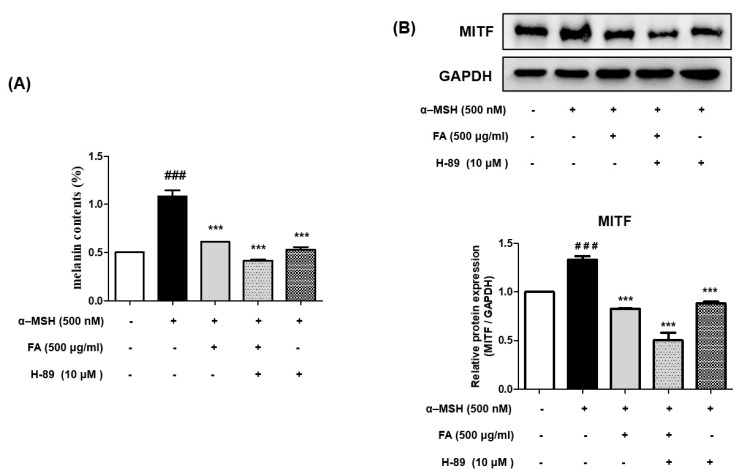
(**A**) Effect of FA and H-89 (PKA/CREB inhibitor) on melanin content (%) in α-MSH-treated B16F10 cells after 48 h. Each value is presented as the mean ± SD. ### *p* < 0.001 (significant difference versus control); *** *p* < 0.001 (significant difference versus α-MSH-treated group). (**B**) Effects of FA and H-89 (PKA/CREB inhibitor) on protein expression of MITF in α-MSH-treated B16F10 cells after 48 h. Each value is presented as the mean ± SD. ### *p* < 0.001 (significant difference versus control); *** *p* < 0.001 (significant difference versus α-MSH-treated group).

## Data Availability

All data and information about the materials and methods in this study are available from the corresponding author upon request.

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
