# Peer review of "Fermented Aronia melanocarpa Inhibits Melanogenesis through Dual Mechanisms of the PI3K/AKT/GSK-3β and PKA/CREB Pathways"

_molecules, 2023, doi:10.3390/molecules28072981_

Round 1

Reviewer 1 Report

The Introduction section is difficult to read and understand due to the lack of schemes, that explain the interactions between the signalling pathways mentioned in the text.

The main active molecule of FA is not clearly detected, only HPLC analysis was performed without NMR of mass-analysis, it is not enough for compound identification. 

It is not applicable to use different types of concentration in determination of activity.

Gallic acid should be used as positive control in all experiments, including antiproliferative activity

Is there any data concerning other components of FA? Why authors suggest only gallic acid to exhibit biological activity? FA may contain other active compounds

The concentration of gallic acid in FA and UA should be determined

Author Response

The Introduction section is difficult to read and understand due to the lack of schemes, that explain the interactions between the signalling pathways mentioned in the text.

Response) Thank you for your feedback. We have added the explanation of the signaling pathway, which we performed in our experiments.

The main active molecule of FA is not clearly detected, only HPLC analysis was performed without NMR of mass-analysis, it is not enough for compound identification. 

Response) The reason for using HPLC analysis was to find out which substances had increased content among the components of fermented Aronia melanocarpa. We selected gallic acid as an indicator material on HPLC based on references that gallic acid affected whitening activity, and confirmed that the peak was higher in the fermented product of Aronia than in the original product.

It is not applicable to use different types of concentration in determination of activity.

Response) Based on references regarding whitening experiments, the concentration of arbutin used as a positive control was 1 mM because it had no cytotoxicity and the maximum whitening activity.

Gallic acid should be used as positive control in all experiments, including antiproliferative activity

Response) You can think so. However, we found gallic acid as an increased product of fermented Aronia melanocarpa. That is, it was not to determine how much whitening activity fermented Aronia melanocarpa compared to gallic acid. Therefore, gallic acid was not used as a positive control group, and arbutin was used as a positive control group. We have investigated the gallic acid activities as well in α-MSH induced B16F10 cells.

Is there any data concerning other components of FA? Why authors suggest only gallic acid to exhibit biological activity? FA may contain other active compounds

Response) As a result of FA content analysis using HPLC, we confirmed that there were flavonoids including anthocyanins. However, as we focused our research on gallic acid, we ask for your understanding of the fact that we paid less attention.

The concentration of gallic acid in FA and UA should be determined

Response) Our study is focused on identifying the major bioactive compound in Aronia melanocarpa, and their bio-functional activities against melanogenesis. Therefore, we detected the gallic acid in UA and FA rather than quantitative analysis of gallic acid in UA and FA.

Reviewer 2 Report

The manuscript of D. H. Kim, D. W. Shin, and B. O. Lim  “Fermented Aronia melanocarpa Inhibits Melanogenesis through Dual Mechanisms of the PI3K/AKT/ GSK-3β and 3 PKA/CREB Pathway” summarizes experimental data on the effect of fermented extract from the fruits of Aronia melanocarpa on the process of formation of melanin in B16F10 melanoma cells. The manuscript contains a lot of experimental data obtained by various biochemical methods. The text of the manuscript contains a certain logic of research. However, unfortunately, both in the text of the article and in the presentation of some data there are some inaccuracies that make it difficult to evaluate this study.

The main comment refers to the concept of the article itself. In the text of the article, the Authors mention several times that such a study was conducted for the first time (lines 79-80: This is the first study as it has never been conducted before. Lines 172-173: The key concept of this study is that M. purpureus fermentation can increase the content of polyphenols or flavonoids in the fermentation substrate. Lnes 292-293: our work is the first study to report on the anti-melanin activity of 292 Aronia melanocarpa.). At the same time, it remains unclear exactly what was done for the first time. Based on previous studies, including those to which the Authors refer, it becomes clear that the content of the main biologically active substances in extracts of Aronia is known, it is known that the relative content of polyphenols increases after fermentation, and the effect of polyphenols, in particular, gallic acid, on melanogenesis is also known. In my opinion, the novelty of this study consists in obtaining a fermented extract using M. purpureus and studying its effect on various signaling pathways in cells that lead to the formation of melanin. The authors should formulate the novelty of this study more clearly.

Another comment. The study compares the effects of fermented extract (FA) and non-fermented (UA). It seems to me important to compare the composition of these extracts and clearly and quantitatively describe the content of active ingredients in both extracts. Otherwise, it is completely unclear what the Authors are working with. This should be described before the results of biochemical experiments (now this information is done in the middle of the text in section 2.6.)

There are also comments to section 2.6 and Figure 6. Obviously, the gradient in which the separation is shown in Figure 6 and which is described in Section 4.5 is not optimal: gallic acid comes out in this gradient at 4 min (Figure 6). It would be possible to use a gradient not from 15% of acetonitrile, but from 0 or 5% (for example). As for the Figure 6, it is not necessary to depict the entire process of washing and balancing the column on it, but only the gradient itself (during 20 min) or a fragment of it where the peaks come out.

It is surprising that only gallic acid is present in the extracts on the HPLC chromatogram (at 270 or 280 nm detection - different sections indicate differently). Either the Authors need to apply a different gradient, and/or detection at different wavelengths. Also, in my opinion, there is not enough HPLC data to determine the composition of extracts, it would be necessary to use the LC-MS method or an another MS-method.

Summing up this comment, the extracts should be more carefully characterized.

Other comments

Line 41. …….tyrosinase-related protein 1, 2 (TRP1,2) – should be …..tyrosinase-related protein 1 and 2 (TRP-1 and TRP-2)

An abbreviation UA appears in the text on line 84 (Results, 2.1) and further, and what this abbreviation means is given only in the caption to Figure 6 (!) and in section 4.4.

Lines 90-91. ……cells were treated with UA 90 and FA at 500 μg/mL, a concentration within the range of 50 μg/mL to 800 μg/ml. - What was the concentration?

Line 101. …… inhibited α-MSH-induced melanin synthesis, but melanin synthesis (Fig. 2). - an incomprehensible phrase.

Figure 2 - there is no caption for panel A.

There is an abbreviation TYR for tyrosinase in the text, however, both the full name and the abbreviation are used at the same time.

Figure. The Figures use fragments of gels. Full imaging of gels also should be shown in the Supplementary materials.

Lines 310-311. To understand the mechanism of the anti-melanogenic action of FA in detail, the effect of the compound on the melanogenesis-related enzymes was evaluated. – What compound?

Figures 4 and 5 captions. Each value was…….. – is?

Section 4.4. It is unclear whether fermentation was performed for the first time using M. purpureus. Is the procedure known? There are no references to the procedure.

Section 4.5. Column – 5 m or 5 micro m? Inj. volume – 10 L or 10 microL? Vavelength – 280 or 270 nm?

Line 380. Gradient instead gadients.

Not all abbreviations in the text are deciphered.

The manuscript also needs careful editing and improvement of English.

The article can be accepted for publication, but needs significant revision.

Author Response

The manuscript of D. H. Kim, D. W. Shin, and B. O. Lim  “Fermented Aronia melanocarpa Inhibits Melanogenesis through Dual Mechanisms of the PI3K/AKT/ GSK-3β and 3 PKA/CREB Pathway” summarizes experimental data on the effect of fermented extract from the fruits of Aronia melanocarpa on the process of formation of melanin in B16F10 melanoma cells. The manuscript contains a lot of experimental data obtained by various biochemical methods. The text of the manuscript contains a certain logic of research. However, unfortunately, both in the text of the article and in the presentation of some data there are some inaccuracies that make it difficult to evaluate this study.

The main comment refers to the concept of the article itself. In the text of the article, the Authors mention several times that such a study was conducted for the first time (lines 79-80: This is the first study as it has never been conducted before. Lines 172-173: The key concept of this study is that M. purpureus fermentation can increase the content of polyphenols or flavonoids in the fermentation substrate. Lines 292-293: our work is the first study to report on the anti-melanin activity of 292 Aronia melanocarpa.). At the same time, it remains unclear exactly what was done for the first time. Based on previous studies, including those to which the Authors refer, it becomes clear that the content of the main biologically active substances in extracts of Aronia is known, it is known that the relative content of polyphenols increases after fermentation, and the effect of polyphenols, in particular, gallic acid, on melanogenesis is also known. In my opinion, the novelty of this study consists in obtaining a fermented extract using M. purpureus and studying its effect on various signaling pathways in cells that lead to the formation of melanin. The authors should formulate the novelty of this study more clearly.

Response) We agree with your feedback. Our subject was not the whitening activity of gallic acid. We demonstrated that the content of gallic acid in Aronia melanocarpa fermented with Monascus purpureus was increased and that FA exhibited more whitening activity compared to UA. That is, it is not simply data that gallic acid has whitening activity.

Another comment. The study compares the effects of fermented extract (FA) and non-fermented (UA). It seems to me important to compare the composition of these extracts and clearly and quantitatively describe the content of active ingredients in both extracts. Otherwise, it is completely unclear what the Authors are working with. This should be described before the results of biochemical experiments (now this information is done in the middle of the text in section 2.6.)

Response) Thank you for your feedback. The reason for using HPLC analysis was to find out which substances had increased content among the components of fermented Aronia melanocarpa. We selected gallic acid as an indicator material on HPLC based on references that gallic acid had an effect on whitening activity, and confirmed that the peak was higher in the fermented product of Aronia than in the original product.

There are also comments to section 2.6 and Figure 6. Obviously, the gradient in which the separation is shown in Figure 6 and which is described in Section 4.5 is not optimal: gallic acid comes out in this gradient at 4 min (Figure 6). It would be possible to use a gradient not from 15% of acetonitrile, but from 0 or 5% (for example). As for the Figure 6, it is not necessary to depict the entire process of washing and balancing the column on it, but only the gradient itself (during 20 min) or a fragment of it where the peaks come out.

Response) A similar retention time of gallic acid was exerted in at previous study (https://doi.org/10.1080/09540105.2020.1742667), where the use gradient acetonitrile from 0% at the beginning.

It is surprising that only gallic acid is present in the extracts on the HPLC chromatogram (at 270 or 280 nm detection - different sections indicate differently). Either the Authors need to apply a different gradient, and/or detection at different wavelengths. Also, in my opinion, there is not enough HPLC data to determine the composition of extracts, it would be necessary to use the LC-MS method or an another MS-method.

Response) Aronia has been reported high content of anthocyanin, flavonoids, and other polyphenols (https://doi.org/10.3390/molecules23030615 and https://www.ncbi.nlm.nih.gov/pmc/articles/PMC8747965/). Hence, our study is focused on identifying the major bioactive compound in Aronia and their bio-functional activities against melanogenesis. Therefore, we detected the gallic acid in UA and FA rather than quantitative analysis of gallic acid in UA and FA.

Summing up this comment, the extracts should be more carefully characterized.

Other comments

Line 41. …….tyrosinase-related protein 1, 2 (TRP1,2) – should be …..tyrosinase-related protein 1 and 2 (TRP-1 and TRP-2)

Response) Thank you for your feedback. We fixed it.

An abbreviation UA appears in the text on line 84 (Results, 2.1) and further, and what this abbreviation means is given only in the caption to Figure 6 (!) and in section 4.4.

Response) Thank you for your feedback. We fixed it.

Lines 90-91. ……cells were treated with UA 90 and FA at 500 μg/mL, a concentration within the range of 50 μg/mL to 800 μg/ml. - What was the concentration?

Response) Thank you for your feedback. We deleted the sentence “a concentration within the range of 50 μg/mL to 800 μg/ml.”

Line 101. …… inhibited α-MSH-induced melanin synthesis, but melanin synthesis (Fig. 2). - an incomprehensible phrase.

Response) Thank you for your feedback. We deleted the sentence “, but melanin synthesis”

Figure 2 - there is no caption for panel A.

Response) Thank you for your feedback. We deleted the caption for the panel A

There is an abbreviation TYR for tyrosinase in the text, however, both the full name and the abbreviation are used at the same time.

Response) Thank you for your feedback. We decided to use only tyrosinase.

Figure. The Figures use fragments of gels. Full imaging of gels also should be shown in the Supplementary materials.

Response) Thank you for your feedback. In order to see various proteins at once, cutting by molecular weight was inevitable.

Tyrosinase (60 kDa)/Trp-1/Trp-2 (75 kDa) was cut at the PVDF membrane

MITF (59 kDa)/GAPDH (36kDa) was cut at the PVDF membrane

Lines 310-311. To understand the mechanism of the anti-melanogenic action of FA in detail, the effect of the compound on the melanogenesis-related enzymes was evaluated. – What compound?

Response) Thank you for your feedback. We deleted the wrong sentence “in detail, the effect of the compound on.”

Figures 4 and 5 captions. Each value was…….. – is?

Response) Thank you for your feedback. We changed into “Each value of UA, FA and arbutin ~”

Section 4.4. It is unclear whether fermentation was performed for the first time using M. purpureus. Is the procedure known? There are no references to the procedure.

Response) We could not find any papers related to fermented Aronia melanocarpa by using M. purpureus

Section 4.5. Column – 5 m or 5 micro m? Inj. volume – 10 L or 10 microL? Wavelength – 280 or 270 nm?

Response) Thank you for your feedback. We fixed those. 5 um particle size, Inj. Volume is 10 uL, the wavelength is 280 nm

Line 380. Gradient instead gadients.

Response) We already wrote gradients in section 4.4.

Not all abbreviations in the text are deciphered.

Response) We carefully checked and fixed them.

The manuscript also needs careful editing and improvement of English.

Response) We asked the English editing service, and we will submit the certificate in the revised manuscript.

The article can be accepted for publication, but needs significant revision.

Response) Thank you for your feedback.

Reviewer 3 Report

The article describes additional material in the fermentation of AM and its pharmacological efficacy against melanogenesis through multiple mechanisms. While overall the article is not ss bad, the results are interesting and the supporting western blots data further enhance the quality of results but still there are some outcoming which needed to be properly addressed.

1. The title, abstract and paper body is totally different, the title and abstract should be more explanatory, especially the abstract, I strongly recommend to properly discussing the abstract. An abstract is very confusing, not matching the article's body. line 365. "Ethanol was removed from the extract using a vacuum rotary concentrator" It should be "The ethanolic extract was dried using a vacuum rotary concentrator". not ethanol was removed". 

LIne 374-384. Why there are many capital alphabets in mid of the sentences???

Line 379. "an injection volume of 10 L" Are you serious??? the injection volume is 10L???????

2. I suggest authors provide colorful graphs instead of black and white.

3. The HPLC chromatograms are not clear enough I cannot see anything in the chromatogram. Provide the high-quality figures.

4. English writing of the paper is very poor. In the abstract, the authors mentioned "natural plants" I am wondered is there anyone using artificial plants for research of secondary metabolites.

5. There are many things that can enhance the production of polyphenols, why the authors only used Monascus? What if authors tried something else and would have better results than monsacuss?

6. Conclusion is missing.

7. The references are very old. It should be properly updated.

Author Response

The article describes additional material in the fermentation of AM and its pharmacological efficacy against melanogenesis through multiple mechanisms. While overall the article is not so bad, the results are interesting and the supporting western blots data further enhance the quality of results but still there are some outcoming which needed to be properly addressed.

1. The title, abstract and paper body is totally different, the title and abstract should be more explanatory, especially the abstract, I strongly recommend to properly discussing the abstract. An abstract is very confusing, not matching the article's body.

Response) Thank you for your feedback. We tried to follow your feedback.

line 365. "Ethanol was removed from the extract using a vacuum rotary concentrator" It should be "The ethanolic extract was dried using a vacuum rotary concentrator". not ethanol was removed". 

Response) Thank you for your feedback. We fixed it as you mentioned. 

LIne 374-384. Why there are many capital alphabets in mid of the sentences???

Response) Thank you for your feedback. We fixed all.

Line 379. "an injection volume of 10 L" Are you serious??? the injection volume is 10L???????

Response) Thank you for your feedback. We made a mistake, and fixed 10 μL

2. I suggest authors provide colorful graphs instead of black and white.

Response) Thank you for your feedback. This is a sort of Authors’ preference. Please, understand that we never used the color graph in our previous papers.

3. The HPLC chromatograms are not clear enough I cannot see anything in the chromatogram. Provide the high-quality figures.

Response) Thank you for your feedback. We will provide the high-quality figures.

4. English writing of the paper is very poor. In the abstract, the authors mentioned "natural plants" I am wondered is there anyone using artificial plants for research of secondary metabolites.

Response) Thank you for your feedback. We asked the English editing service, and we will submit the certificate in the revised manuscript.

5. There are many things that can enhance the production of polyphenols, why the authors only used Monascus? What if authors tried something else and would have better results than monsacuss?

Response) Yes, we agree with your feedback. There are many ways to increase the polyphenol content. We used it because it is a fermentation strain currently set in our laboratory, and other results are expected through sufficiently different fermentation strains. In our study, we emphasized that Aronia melanocarpa fermented using M. purpureus is the first manuscript.

6. Conclusion is missing.

Response) Thank you for your feedback. The format of “Molecules” has a Discussion, which includes a Conclusion.

7. The references are very old. It should be properly updated.

Response) Thank you for your feedback. We changed and added new references.

Round 2

Reviewer 1 Report

Authors solved major issues in revised manuscript, so it can be accepted

Author Response

Dear Reviewer

Authors solved major issues in revised manuscript, so it can be accepted

Answer) Thank you so much. Owing to your good feedback, our manuscript must be improved.  

Reviewer 2 Report

The Authors have made some corrections to the text of the manuscript, which make the text and goals more logical and specific. Also, the Authors responded to the comments of the Reviewers. However, the comments remain and some corrections and clarifications need to be made in order for the article to be published.

Lines 74 and following. UA and FA. Be sure to specify the full term, and then enter abbreviations! (I repeat this comment)

Lines 101-102. ….the inhibitory effect of FA on melanin synthesis was significantly better….. – stronger or weaker?

Figure 2. The left panel of the drawing (photo on the left) is not signed (I repeat this comment)

Figures 3-9. The quality of the Figures is low. It is necessary to present the Figures in a higher quality.

Legends (Figures 3-4). Each value of UA, FA, and arbutin is presented as….. - What values are given? Concentration, dose?

Figure 6. All comments about Figure 6 remain. The Authors answered that they used the gradient applied earlier and provided a reference. But, firstly, common sense requires applying a more suitable gradient (especially since it is described). Secondly, Figure 6 is extremely uninformative and of very poor quality. It is required to redo this experiment so that the composition of extracts and the retention time of gallic acid are clear!

Line 378. Ethanol was dried from the extract using a vacuum rotary concentrator – should be: Ethanol was evaporated from…….

Authors should carefully check the text of the manuscript for such inaccuracies. It is desirable that the text be checked by native English speakers.

Author Response

The Authors have made some corrections to the text of the manuscript, which make the text and goals more logical and specific.

Also, the Authors responded to the comments of the Reviewers. 

Response) Thank you for your comments

However, the comments remain and some corrections and clarifications need to be made in order for the article to be published.

Lines 74 and following. UA and FA. Be sure to specify the full term, and then enter abbreviations! (I repeat this comment)

Response)  Thank you for your comment. We added the full term “unfermented Aronia melanocarpa (UA) and fermented Aronia melanocarpa (FA).”

Lines 101-102. ….the inhibitory effect of FA on melanin synthesis was significantly better….. – stronger or weaker?

Response) Thank you for your comment. We changed into “stronger” by considering Figure 2.

Figure 2. The left panel of the drawing (photo on the left) is not signed (I repeat this comment)

Response) Thank you for your comment. We rechanged the figure image.

Figures 3-9. The quality of the Figures is low. It is necessary to present the Figures in a higher quality.

Response) Thank you for your comment. Thus, we replaced all figure images with JPEG rather than ppt and amplified all figures compared to the previous figures.

Legends (Figures 3-4). Each value of UA, FA, and arbutin is presented as….. - What values are given? Concentration, dose?

Response) Thank you for your feedback. We deleted “UA, FA, and arbutin”, which are meaningless.

Figure 6. All comments about Figure 6 remain. The Authors answered that they used the gradient applied earlier and provided a reference. But, firstly, common sense requires applying a more suitable gradient (especially since it is described). Secondly, Figure 6 is extremely uninformative and of very poor quality. It is required to redo this experiment so that the composition of extracts and the retention time of gallic acid are clear!

Response) Thank you for your feedback. We agreed with your opinion. A similar retention time of gallic acid was exerted in a previous study (https://doi.org/10.1080/09540105.2020.1742667), where the use of gradient acetonitrile was from 0% at the beginning. Our study is focused on identifying the major bioactive compound in Aronia and their bio-functional activities against melanogenesis. Therefore, we detected the gallic acid in UA and FA rather than quantitative analysis of gallic acid in UA and FA. Unfortunately, we are sorry that our experimental environment was insufficient to give additional information except for the difference in the gallic acid content in both FA and UA.

Line 378. Ethanol was dried from the extract using a vacuum rotary concentrator – should be: Ethanol was evaporated from…….

Response) Reviewer 1 asked us to correct “dried”. But the word using “evaporated” is better than “dried”.

Authors should carefully check the text of the manuscript for such inaccuracies. It is desirable that the text be checked by native English speakers.

Response) Thank you for your comments. We asked the MDPI English editing service for our revised manuscript.

Reviewer 3 Report

The authors addressed all my questions very clearly. I recommend for its publication.

Author Response

Dear Reviewer 

The authors addressed all my questions very clearly. I recommend for its publication.

Answer) Thank you so much. Owing to your good review, our revised manuscript must be improved.